# Opioid Prescription Method for Breathlessness Due to Non-Cancer Chronic Respiratory Diseases: A Systematic Review

**DOI:** 10.3390/ijerph19084907

**Published:** 2022-04-18

**Authors:** Yasuhiro Yamaguchi, K.M. Saif-Ur-Rahman, Motoko Nomura, Hiromitsu Ohta, Yoshihisa Hirakawa, Takashi Yamanaka, Satoshi Hirahara, Hisayuki Miura

**Affiliations:** 1Department of Pulmonary Medicine, Jichi Medical University Saitama Medical Center, Saitama 330-8503, Japan; motoko862@jichi.ac.jp (M.N.); hootatky@jichi.ac.jp (H.O.); 2Department of Public Health and Health Systems, Graduate School of Medicine, Nagoya University, Nagoya 466-8550, Japan; s.rahman2312@gmail.com (K.S.-U.-R.); y.hirakawa19710718@gmail.com (Y.H.); 3Health Systems and Population Studies Division, ICDDR,B, Dhaka 1212, Bangladesh; 4Department of Home Care Medicine, Graduate School of Medicine, The University of Tokyo, Tokyo 113-8655, Japan; yamanaka.takashi@mail.u-tokyo.ac.jp; 5Tokyo Fureai Medical Co-op Research & Education Center, Tokyo 114-0004, Japan; hirahara@fureaico-op.com; 6Department of Home Care and Regional Liaison Promotion, National Center for Geriatrics and Gerontology, Obu 474-8511, Japan; hmiura@ncgg.go.jp

**Keywords:** breathlessness, chronic obstructive pulmonary disease, dyspnea, interstitial lung disease, morphine, opioid, palliative care

## Abstract

A previous pooled analysis demonstrated significant relief of breathlessness following opioid administration in patients with chronic obstructive pulmonary disease. However, in clinical practice, it is important to know the characteristics of patients responding to opioids, the best prescription methods, and the evaluation measures that can sufficiently reflect these effects. Thus, we performed a systematic review of systemic opioids for non-cancer chronic respiratory diseases. Fifteen randomized controlled studies (RCTs), four non-randomized studies, two observational studies, and five retrospective studies were included. Recent RCTs suggested that regular oral opioid use would decrease the worst breathlessness in patients with a modified Medical Research Council score ≥ 3 by a degree of 1.0 or less on a scale of 1–10. Ergometer or treadmill tests indicated mostly consistent significant acute effects of morphine or codeine. In two non-randomized studies, about 60% of patients responded to opioids and showed definite improvement in symptoms and quality of life. Furthermore, titration of opioids in these studies suggested that a major proportion of these responders had benefits after administration of approximately 10 mg/day of morphine. However, more studies are needed to clarify the prescription method to reduce withdrawal due to adverse effects, which would lead to significant improvements in overall well-being.

## 1. Introduction

A recent global survey found that chronic respiratory diseases were the third leading cause of death in 2017 [1]. Furthermore, advanced chronic respiratory diseases impair quality of life (QoL) as severely as cancer before death [2]. Improvement in palliative care is therefore an important issue for these diseases.

Breathlessness is one of the most frequent symptoms of advanced chronic respiratory diseases [3,4,5]. Management of the causal diseases is necessary for the relief of dyspnea, such as inhaled steroids for asthma and bronchodilators for chronic obstructive pulmonary disease (COPD). Pulmonary rehabilitation is also available to improve exercise tolerance and relieve mental distress in advanced COPD [6,7]. In addition to these measures, systemic opioids are an important alternative for intractable breathlessness [8,9]. A previous pooled analysis revealed a significant improvement in breathlessness due to opioids in patients with COPD [10].

However, recent randomized controlled trials (RCTs) did not demonstrate any significant differences in many outcomes between opioids and placebo [11,12]. The inconsistency of these results could be a barrier to prescribing opioids for breathlessness in cases of non-cancer disease [13]. In Japan, opioids for dyspnea have not been approved by the national insurance system, possibly owing to the absence of clear evidence of their efficacy. To establish the evidence of opioids as a treatment for breathlessness, the cause of these inconsistencies should be clarified systematically, and the subgroups in which many patients do respond well to opioids should be demonstrated.

In clinical practice, physicians and patients experience difficulty in palliating breathlessness when chronic respiratory diseases, such as COPD and interstitial lung disease (ILD), advance to breathlessness corresponding to a modified Medical Research Council (mMRC) score ≥ 3 or 4. The expected efficacy of opioids should be determined for these patients. Simultaneously, both physicians and patients with non-cancer diseases prioritize the prevention of adverse events that can affect daily activities or prognosis. The safest prescription methods should be pursued in order to break the barriers that prevent the use of opioids for the palliation of breathlessness in non-cancer diseases.

For these reasons, we performed a systematic review of articles that evaluated the effects of opioids in advanced chronic respiratory diseases.

## 2. Methods

This systematic review was performed in accordance with the Preferred Reporting Items for Systematic Reviews and Meta-Analyses guidelines [14]. The study protocol was registered in the PROSPERO database (CRD42021267919).

### 2.1. Data Sources

We searched and identified articles described in English in MEDLINE (from 1980 to October 2020) and the Cochrane Library (from 1980 to October 2020). The search strategy is provided in the online Appendix A.

### 2.2. Study Selection

We searched for studies that evaluated the effect of oral or parenteral opioids as interventions, excluding nebulized opioids. The participants had respiratory symptoms, primarily due to chronic respiratory diseases other than cancer. We included studies that partially involved participants with chronic diseases of other organs or malignant tumors. The outcomes included clinical changes in respiratory symptoms, health-related QoL, and exercise tolerance after opioid administration. The study types were RCTs, non-randomized, observational, and retrospective studies involving >5 patients.

We excluded studies that exclusively involved respiratory symptoms due to cancer or chronic diseases of other organs, such as cardiac and neurological diseases. We also excluded studies that exclusively focused on the adverse effects of opioids or opioid prescription rates. Qualitative studies were also excluded.

YY and KMSUR screened the titles and abstracts of all studies identified by the search strategy and performed full-text assessments to identify studies for inclusion. Disagreements were resolved by discussion with a third reviewer, HO, or MN. Referring to the studies selected in previous systematic reviews, YY assessed for disagreements with our study selection. Regarding the studies that were not selected in our search strategy, at least two review authors evaluated eligibility for inclusion as hand-searched studies.

### 2.3. Data Extraction and Analysis

We extracted data on background respiratory diseases, baseline dyspnea level, opioid classifications, opioid dosage, duration of the intervention, breathlessness after the intervention, exercise tolerance after the intervention, QoL after the intervention, and adverse effects of opioids. Data extraction was first performed by a reviewer and then checked by another reviewer.

Data are presented descriptively. As the studies had significant heterogeneity and data varied between studies, we performed a narrative review to assess the differences in response to opioids among various severities of dyspnea, evaluation measures, and opioid prescriptions.

### 2.4. Risk of Bias Assessment

As for RCTs, risk of bias assessment was performed according to the Cochrane Risk of Bias (ROB) tool [15]. For studies other than RCTs, the Risk of Bias Assessment tool for Non-randomized Studies (RoBANS) was used [16]. Two review authors independently assessed the risk of bias of the included studies and any discrepancies were resolved through discussion. The domains considered at ROB were random sequence generation, allocation concealment, blinding of participants and personnel, blinding of outcome assessors, incomplete outcome data, selective reporting, and other bias, whereas the RoBANS tool considered selection of participants, confounding variables, measurement of exposure, blinding of outcome assessment, incomplete outcome data, and selective outcome reporting.

## 3. Results

### 3.1. Characteristics of Selected Studies

A total of 713 publications were identified in the initial search, of which 30 were suitable for inclusion in the systematic review. We also referenced two previous systematic reviews on opioids in respiratory diseases [10,17], and two studies [18,19] that were not selected through our database search were included. Therefore, 32 articles were included in this systematic review (Figure 1).

Among the five articles [20,21,22,23,24] that demonstrated sub-analysis of the previous studies [11,25,26], four [20,22,23,24] did not provide any additional information on breathlessness, QoL, or exercise tolerance and were excluded from further review. The article by Smith J. et al. [27] evaluated the effect of opioids exclusively on coughing and was excluded from further review. The article by Ferreira D.H. et al. [28] was the only study that evaluated pulmonary arterial hypertension, in which morphine did not reduce breathlessness. Because pulmonary arterial hypertension would differ in the pathogenesis of breathlessness from other chronic respiratory diseases, this study was also excluded from further review. The characteristics of the remaining 15 RCTs are shown in Table 1, Table 2 and Table 3. Further to the studies already mentioned, there were four non-randomized studies (Table 4 and Table 5), two observational studies (Table 4 and Table 5), and five retrospective studies (Table 6).

### 3.2. RCT Studies of Opioids to Reduce Breathlessness Due to COPD

Two relatively large-scale RCTs have recently reported no significant differences in the clinical scores of breathlessness between opioids and placebo [11,12]. The sub-analysis of patients with mMRC score ≥ 3 demonstrated that the numerical rating scale (NRS) or visual analog scale (VAS) of the worst breathlessness was significantly reduced in patients who received opioids compared to those who received placebo in both studies.

There were four other RCTs that recruited patients with dyspnea corresponding to mMRC ≥ 3. Abdallah S.J. et al. [34] showed that morphine reduced exertional breathlessness in cardiopulmonary cycle exercise testing (Table 3). Woodcock A.A. et al. [18] also demonstrated that dihydrocodeine significantly improved subjective disability by measuring the oxygen cost diagram, although the attrition bias in this study was high owing to the high percentage of withdrawal after taking dihydrocodeine due to side effects (Table 1 and Table 2). A crossover study conducted by Abernethy A.P. et al. [25] was the only RCT that recruited patients with breathlessness at rest and indicated that the VAS score of breathlessness was significantly reduced under sustained-release (SR) morphine compared to placebo (Table 1 and Table 2). Contrary to the consistent effectiveness of morphine or dihydrocodeine for dyspnea of mMRC ≥ 3, the study by Ferreira D.H. et al. [21], which is derived from the immaturely ceased portion of the same trial as that by Currow D. et al. [12], did not demonstrate any difference in breathlessness scores between oxycodone and placebo, including VAS score of the worst breathlessness in the previous 24 h. No significant improvement in breathlessness was shown by regularly used opioids in any studies which included patients with dyspnea scores corresponding to a mMRC of 2, or those which had no criteria of enrollment regarding dyspnea (Table 1 and Table 2).

Notably, the average degree of improvement in breathlessness caused by opioids was low. The difference in the worst breathlessness in the previous 24 h was −1.33 (95% confidence interval, −2.50, −0.16) on the NRS (0−10) in COPD patients with dyspnea of mMRC ≥ 3 in the study by Verberkt C.A. et al. [11]. The differences in other studies [12,21,25,29,31,33,36] were even smaller. Studies that used rougher scales, such as the studies by Poole PJ et al. [30] and Munck L.K. et al. [32], did not detect significant differences in breathlessness between opioids and placebo.

### 3.3. RCT Studies of Opioids to Improve Exercise Tolerance

There were no RCT studies of regularly used opioids which indicated significant differences in exercise tolerance favoring opioids (Table 1) [11,18,29,30,31,33]. Poole P.J. et al. [30] indicated that the distance in the 6MWT was significantly shorter under morphine. In contrast to these negative results, five of the seven studies on the acute effects of opioids [19,29,31,34,35,36,37] indicated significantly better results, except for the study of diamorphine using the 6MWT by Eiser N. et al. [31] and the small study by Light R.W. et al. [19] (Table 3). The study by Johnson M.A. et al. [36] also provided instructions to consume a tablet of dihydrocodeine or placebo 30 min before exercise, and the mean number of tablets used during the dihydrocodeine week was 2.8 tablets, possibly reflecting an acute effect.

The evaluation measure is another factor that can reflect the effectiveness of opioids in exercise tolerance. Six studies [11,18,29,30,31,33] evaluated exercise tolerance by using a 6 min walk test (6MWT) or 12 min walk test (12MWT). There were no significant differences in the 6MWT or 12MWT distances favoring opioids compared with the placebo groups. While the 6MWT results by Kronborg-White S. et al. [29] indicated significantly better changes on the Borg scale 1 h after the first immediate-release (IR) morphine intake, there were no other studies that indicated differences in breathlessness after 6MWT between groups that received opioids and placebo [30,31]. In contrast, four of the six studies that evaluated exercise tolerance using a cycle ergometer or treadmill [19,31,34,35,36,37] showed significantly better results in patients taking opioids [34,35,36,37].

Only one study has evaluated the effect of regularly used opioids on exercise tolerance using a cycle ergometer or treadmill, which demonstrated no significant differences in exercise tolerance or breathlessness between diamorphine and placebo [31].

### 3.4. RCT Studies of Opioids to Improve QOL in COPD

Verberkt C.A. et al. [11] indicated that COPD assessment test scores were significantly reduced in patients who received morphine compared to those who received placebo, favoring morphine. However, no other studies [12,21,29,30] have indicated significant differences in QoL between opioid and placebo groups (Table 5). A study by Poole P.J. et al. [30] indicated a significant worsening of chronic respiratory questionnaire (CRQ) mastery and a non-significant improvement in CRQ dyspnea in the morphine group.

### 3.5. The Response Rate of Opioids in RCT Studies

It is important in clinical practice to know the percentage of patients with dyspnea who prefer opioids. However, only a few studies have addressed this issue. Among the studies of patients with mMRC ≥ 3, Abdallah S.J. et al. [34] demonstrated that 75% preferred morphine over a placebo during the exercise test (Table 7). In the study by Verberkt C.A. et al. [11], 48% of the patients with mMRC ≥ 2 in the morphine group showed an improvement of ≥1 NRS, which was not significantly different from that of the placebo group (35%) (Table 2). The study by Currow D. et al. [12] demonstrated that the percentage of patients reporting less breathlessness during the past week was not different between opioid and placebo groups (48.5% vs. 49.3%). However, the participants in the placebo group also received rescue morphine in this study (Table 7).

### 3.6. Non-Randomized or Observational Studies of Opioids for Chronic Respiratory Diseases

All four non-randomized studies indicated beneficial effects of opioids in patients with chronic respiratory diseases [26,38,39,40] (Table 4 and Table 5). In a study by Rocker GM et al. [38], 61% of the participants answered that opioid treatment was helpful at 4–6 months. Among these responders, nearly half answered that opioid treatment was very helpful using a 5-point Likert scale. Likewise, 63% of the patients with dyspnea of mMRC ≥ 3 showed a ≥10% reduction in the VAS score of breathlessness without unacceptable side effects in the study by Currow DC et al. [26]. An observational study by Smallwood N et al. [41] also demonstrated that 41.9% self-reported being highly compliant with morphine treatment, although the severity of baseline dyspnea was not indicated in this study (Table 4 and Table 5).

The non-randomized study by Currow DC [26] showed a small improvement in breathlessness in the enrolled patients (Table 5), which was consistent with that reported in RCTs. However, the median NRS score for dyspnea was 2.0 lower after morphine intake than before morphine intake in the patients who continued treatment, that is, the responders to opioids included in the study by Rocker GM et al. [38]. These patients also showed a significant improvement in their CRQ scores (Table 5). Likewise, 52 responders to opioids showed a larger reduction in the VAS score for breathlessness in the study by Currow DC [26] (Table 5).

### 3.7. Retrospective Study of Opioids for Chronic Respiratory Diseases

All five retrospective studies [5,43,44,45,46] indicated beneficial effects of opioids in patients with ILD (Table 6). The studies by Takeyasu M. et al. [43] and Matsuda Y. et al. [44] reported the effect of continuous intravenous infusion or continuous subcutaneous infusion of morphine in advanced ILD.

### 3.8. The Required Dosage of Opioids to Observe Benefits

Five studies determined the oral opioid dosage by titration [11,26,30,38,46]. Among them, the study by Verberkt C.A. et al. [11] started titration from 20 mg/day of morphine, and thus, this study could not determine the effect of a lower dosage. The study by Poole P.J. et al. [30] titrated morphine from 10 mg/day to 40 mg/day if there were no adverse effects or if the adverse effects were minor, and thus, this study could not determine the beneficial effect of a lower dosage (Table 1).

A non-randomized study by Rocker G.M. et al. [38] examined treatment with 2 mg of IR morphine every 4 h during the daytime, and reported that 30% and 45% of participants stated that this was very helpful or somewhat helpful, respectively. Likewise, 69.2% of the responders benefitted from 10 mg/day of SR morphine after titration from 10 mg/day in the study by Currow D. et al. [26]. These findings were consistent with those of the study by Allcroft P. [39], which revealed the beneficial effect of 10 mg/day of morphine with clonazepam (Table 4 and Table 5). However, a retrospective study by Colman R. et al. [46] showed that the median oral morphine equivalent was 30 mg/day in those who were taking SR opioids after titration from a low dose, although a precise titration protocol was not indicated (Table 6).

### 3.9. Adverse Events of Opioid Treatment

There have been no evident reports of severe fatal adverse events. However, the completion rate of the opioid group was approximately 10-20% lower than that of the placebo group in most RCTs regarding regularly used opioids (Table 1). Among eight RCTs, only that by Kronborg-White S. et al. [29] demonstrated a completion rate higher than 90%. Abernethy A.P. et al. [25] also showed that the completion rate of the opioid group was equal to that of the placebo group. In a non-randomized study by Currow DC [26], 18.1% withdrew owing to adverse events. Meanwhile, only 6.8% withdrew owing to adverse events in the study by Rocker G.M. et al. [38].

### 3.10. Risk of Bias of Included Studies

The risk of bias is shown in the online Appendix A. The random sequence generation was mentioned in only six RCTs [11,12,21,25,29,30] and was marked as unclear risk of bias in the rest of the studies. Allocation concealment was not mentioned in most of the RCTs and only three articles were marked as low risk of bias [12,21,25]. Most of the studies mentioned the blinding of participants and personnel, but six studies did not [18,25,30,31,36,37]. Only one article [35] was rated as high risk of bias in this domain. None of the articles mentioned the blinding of outcome assessors and we rated that as unclear risk of bias. Five of the RCTs were rated as high risk of bias for attrition [18,30,31,33,34]. All the RCTs were rated as low risk of bias for selective reporting and other bias.

Similar to the RCTs, blinding of the outcome assessors was not mentioned in any of the case studies, cohort studies, or retrospective studies. All the studies were rated as low risk of bias for performance bias, incomplete outcome data reporting, and selective outcome reporting. Selection of participants was rated as low risk of bias in only three studies [26,41,42], and rated as high risk of bias in the rest of the studies. Only one article did not mention the confounding factors [39], and the rest of the articles were rated as low risk of bias.

## 4. Discussion

In this systematic review, we found consistent findings among heterogeneous studies on opioid use in chronic respiratory diseases. Figure 2 summarizes these findings. Although a previous meta-analysis revealed the beneficial effects of opioids in advanced COPD, it was difficult to elucidate why some studies did not show any improvement in dyspnea due to opioids. Furthermore, previous systematic reviews of opioids for dyspnea did not include recent large-scale RCTs. Additionally, we analyzed non-randomized and retrospective studies because these studies also demonstrated some important findings that could not be indicated in RCTs.

We revealed that the worst breathlessness could be improved by morphine treatment in COPD patients with a mMRC of ≥3 [11,12]. This is consistent with the findings of a previous study that indicated higher baseline breathlessness intensity as a predictor of beneficial responses to opioids [47]. The degree of improvement was also almost same in the two RCTs, approximately 1.0 in 0–10 NRS [11,12]. However, it should be noted that responders to opioids showed clinical improvement more clearly in the two observational studies [26,38]. These findings are important to understand how the effects of opioids should be evaluated in clinical practice.

Regarding the effect on exercise tolerance, the acute effect of morphine or codeine was generally confirmed in ergometer and treadmill studies [34,35,36,37]. However, the 6MWT and 12MWT findings were negative in patients using opioids regularly [11,18,29,30,31,33]. Regrettably, only one study [31] evaluated exercise capacity using a treadmill in patients regularly using opioids. This study used diamorphine but not morphine or codeine. Therefore, it is challenging to determine the differences between the acute and long-term effects of opioids.

Non-randomized studies are also important for determining the percentage of patients responding to opioids and their required dosage in clinical practice [26,38]. Although there is a high risk of the placebo effect, these studies would be appropriately applied to actual practice from the viewpoint of the severity of dyspnea in patients. Likewise, retrospective studies [5,43,44,45] of patients with ILD before death are also important, because no RCT involving these patients has been reported.

Although there was no serious depression of ventilation due to opioids, as indicated in previous systematic reviews [48], the low completion rate of opioid arms in RCTs could be a barrier to recommending opioids in clinical practice. In this systematic review, we could not identify factors that improved the completion rate in some studies [25,29]. Titration from a very low dose of IR morphine to SR morphine would reduce withdrawal due to adverse effects [38], while titration from SR morphine (10 mg) does not prevent adverse effects sufficiently [26,30]. However, other factors, such as the severity of baseline dyspnea, experience with prescribing opioids, and other non-pharmacological support, would also affect the continuation of opioid therapy.

Johnson M.A. et al. [36] also demonstrated relief of breathlessness and a high completion rate without any difference in the occurrence of adverse events from placebo by using dihydrocodeine before exercise. The relatively consistent findings of acute effects of opioids on exercise tolerance would also support the effectiveness of the as-needed use of opioids, although the onset of the effect of IR morphine might be too late for rescue use. No study has evaluated the long-term effects of the as-needed use of IR opioids for chronic respiratory diseases [49,50].

Opioids are currently prescribed for intractable breathlessness in non-cancer respiratory diseases without clear evidence. Although most of our findings were based on a limited number of studies, they provide suggestions for better prescription strategies for opioids in clinical practice. We clearly presented evidence of the effectiveness of morphine or codeine for COPD patients with a mMRC ≥ 3, which can be a cutoff point to consider when initiating opioids. Although adverse events are still a major obstacle to opioid use, the consistent findings regarding the acute effects of opioids on exercise tolerance support the initiation of as-needed low-dose IR morphine or codeine by setting the maximum number of uses. A gradual increment of a very low dose (<10 mg) of morphine would reduce the withdrawal of opioid usage due to adverse events. Nearly 60% responded well to morphine, and the majority of these responders required only low-dose morphine (approximately 10 mg). Although the reports of this response rate were affected by placebo effects, these findings could be directly applied in actual practice.

This systematic review has some limitations. Although some subgroups showed consistent findings in our analysis, the importance of the factors that separated the groups, such as dyspnea severity, opioid classification, acute effects of opioids, and evaluation measures of exercise tolerance, must be confirmed in future prospective studies. Because we could not determine the distribution of dyspnea severity in each study, it is uncertain whether our groupings reflected the actual dyspnea severity of the patients. Additionally, the number of the studies in each subgroup was small. In particular, the absence or small number of applicable studies prevented evaluation of some important issues, such as the effects of opioids other than morphine or codeine, usefulness of the 6MWT for measuring opioid effects, the effect of regularly used opioids on exercise tolerance, and the best prescription method to reduce adverse events.

These limitations could also be related to inconsistency regarding the improvement in QoL by opioids. Further studies are needed to clarify the prescription method to improve overall well-being by using opioids in patients with advanced chronic respiratory diseases.

## 5. Conclusions

Among the heterogeneous studies on opioid use in chronic respiratory diseases, patients with dyspnea corresponding to a mMRC ≥ 3 consistently showed beneficial effects of morphine or codeine. The acute effects of morphine and codeine were also consistently observed in ergometer and treadmill studies. Because a majority of the responders benefitted from low-dose morphine in non-randomized studies, as-needed or regular use of low-dose morphine would be recommended as the first option for severe dyspnea in non-cancer patients. Further studies are needed to clarify the optimal prescription methodology to reduce withdrawal due to adverse effects.

## Figures and Tables

**Figure 1 ijerph-19-04907-f001:**
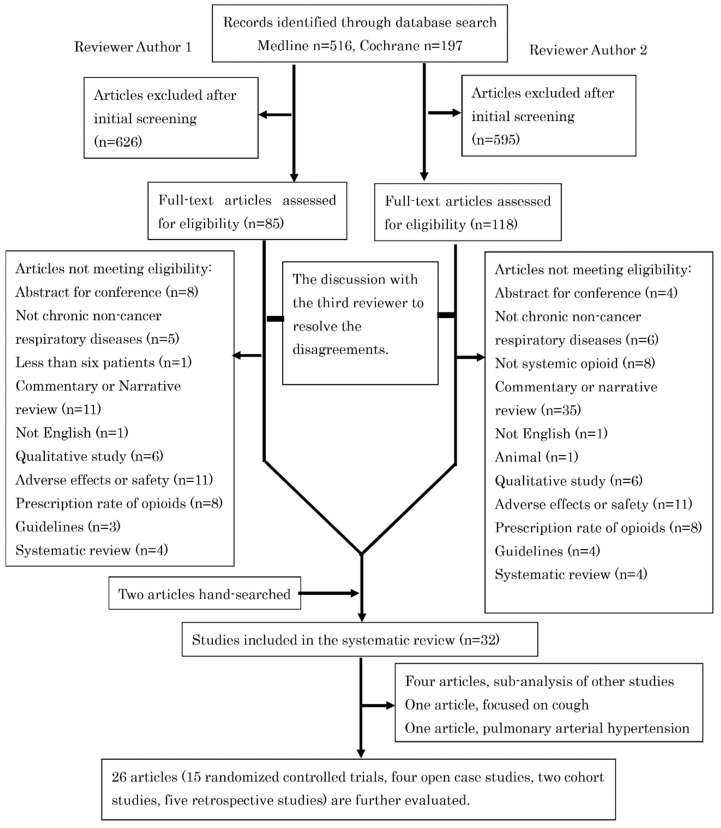
The study selection process of this systematic review.

**Figure 2 ijerph-19-04907-f002:**
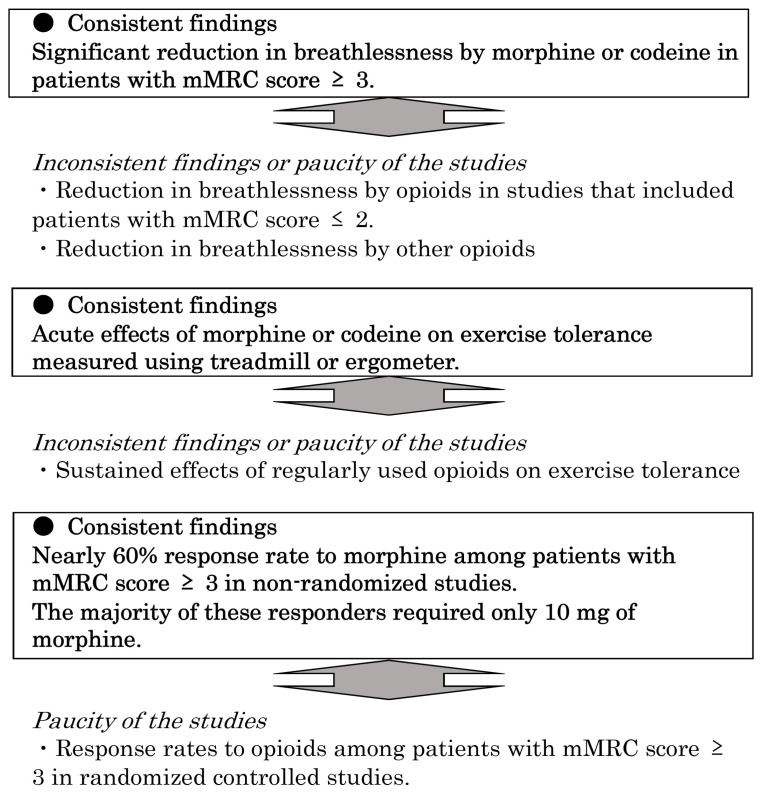
Consistent findings among heterogeneous studies of opioids for breathlessness due to non-cancer chronic respiratory diseases. mMRC, modified Medical Research Council.

**Table 1 ijerph-19-04907-t001:** Characteristics of randomized controlled studies regarding regularly used systemic opioids for non-cancer chronic respiratory disease.

**Parallel Studies**							
**Study**	**Diseases**	**N**	**Completion Rate (%)**	**Criteria of Dyspnea**	**Drug**	**Dose**	**Period**
Verberkt C.A., (2020) [11]	COPD	54 vs. 57	81 vs. 89	mMRC ≥ 2	SR morphine	Titration from 10 mg BID	4 wks
	COPD	23 vs. 26		mMRC ≥ 3	SR morphine	Titration from 10 mg BID	4 wks
Currow D., (2020) [12]	COPD et al.	145 vs. 139	77 vs. 86	mMRC ≥ 2	SR morphine *	20 mg	1 wk
	COPD et al.	88 vs. 79		mMRC ≥ 3	SR morphine *	20 mg	1 wk
Kronborg-White S., (2020) [29]	fILD	18 vs. 18	94.4 vs. 94.4	MRC ≥ 3	IR Morphine	5 mg QID	1 wk
Ferreira D.H., (2019) [21]	COPD et al.	74 vs. 81	73% vs. 85%	mMRC ≥ 3	Oxycodone * 5 mg q8h		1 wk
**Crossover Studies**							
**Study**	**Diseases**	**N**	**Completion Rate (%)**	**Criteria of Dyspnea**	**Drug**	**Dose**	**Period**
Abernethy A.P., (2003) [25]	COPD et al.	48	88.9 vs. 88.6	Dyspnea at rest	SR morphine	20 mg	4 days
Poole P.J., (1998) [30]	COPD	16	87.5 vs. 100	No	SR morphine	Titration from 10 mg	6 wks
Eiser N., (1991) [31]	COPD	18	77.8	No Diamorphine		5.0 mg q6h, 2.5 mg q6h	2 wks
Munck L.K., (1990) [32]	COPD	19	84.2 vs. 100	No	Codeine	60 mg TID	1 wk
Rice K.L., (1987) [33]	COPD	11	70.0 vs. 88.9	No	Codeine	30 mg QID	1 mo
Woodcock A.A., (1982) [18]	COPD	16	68.8 vs. 100	MRC > grade 3	Dihydro-codeine	60 mg TID or 30 mg TID	2 wks

Opioid group vs. placebo group is shown. * Participants in the placebo group also received rescue immediate-release morphine. COPD, chronic obstructive pulmonary disease; fILD, fibrotic interstitial lung diseases; mMRC, modified Medical Research Council; MRC, Medical Research Council; SR, sustained-release; IR, immediate-release; wk, week.

**Table 2 ijerph-19-04907-t002:** Outcomes of randomized controlled studies regarding regularly used systemic opioids for non-cancer chronic respiratory disease.

**Parallel Studies**		
**Study**	**Scores of Breathlessness**	**Exercise Tolerance**
Verberkt C.A., (2020) [11] mMRC ≥ 2	Mean NRS, −0.60 (95%CI, −1.55 to 0.35)	Distance in 6MWT,−5.07 m
	Worst NRS, −0.56 (95%CI, −1.41 to 0.28)	(95% CI, −61.38 to 51.20)
	Improvement of ≥1 NRS, 48% vs. 35%	
Verberkt C.A., (2020) [11] mMRC ≥ 3	Mean NRS, −1.31 (95%CI, −2.80 to 0.17)	Distance in 6MWT, 1.49 m
	Worst NRS, −1.33 (95%CI, −2.50 to −0.16)	(95% CI, −87.47 to 90.46)
Currow D., (2020) [12] mMRC ≥ 2	VAS now, −0.15 (95%CI, −4.59 to 4.29) Average VAS, −2.13 (95%CI, −6.64 to 2.38) Worst VAS, −5.23 (95%CI, −10.77 to 0.31) Dyspnoea (EORTC-QLQ-C15), −1.08 (95%CI, −7.14 to 4.98)	
Currow D., (2020) [12] mMRC ≥ 3	Worst VAS, −7.81 (95%CI, −14.65 to −0.97) No significant differences in other endpoints	
Kronborg-White S., (2020) [29]	Reduction in VAS, −11 ± 14 vs. −3.5 ± 20	Change of distance in 6MWT 10 ± 37 vs. 5 ± 25 Change of Borg scale 0.4 ± 2.5 vs.-0.03 ± 2.1
Ferreira D.H., (2019) [21]	VAS now, 5.33 (95%CI, −1.22 to 11.88) Average VAS, 2.93 (95%CI, −3.08 to 8.95) Worst VAS, −2.51 (95%CI, −10.33 to 5.31) >15% decrease in breathlessness now, 45.6% vs. 50.7% Dyspnoea (EORTC-QLQ-C15), −0.46 (95%CI, −8.60 to 7.67)	
**Crossover Studies**		
**Study**	**Scores of Breathlessness**	**Exercise Tolerance**
Abernethy A.P., (2003) [25]	VAS in morning −6.6 (95%CI, −1.6 to −11.6) VAS in evening −9.5 (95% CI −3.0 to −16.1)	
Poole P.J., (1998) [30]	Daytime breathlessness score score (0–5) 2.22 vs. 2.26 Nighttime breathlessness score score (0–4) 0.81 vs. 0.85 (Higher scores mean worse)	Distance in 6MWT (m), −35.1 ± 70.7 vs. + 21.6 ± 62.1, *p* = 0.04 No difference in 6MWT breathlessness scores
Eiser N., (1991) [31]	Daily VAS, No difference (5mg vs. 2.5 mg vs. placebo)	Distance in 6MWT 226 ± 34 vs. 221 ± 35 vs. 216 ± 40
		VAS dyspnea score for 6MWT, 70 ± 8 vs. 70 ± 7 vs. 65 ± 7 Time on the treadmill (s), 156 ± 47 vs. 104 ± 19 vs. 165 ± 68 VAS dyspnea score for treadmill, 58 ± 10 vs. 66 ± 6 vs. 65 ± 10 (5 mg vs. 2.5 mg vs. placebo)
Munck L.K., (1990) [32]	No difference in dyspnea at rest using NRS 1–4 without detailed data (Codeine + paracetamol vs. paracetamol)	
Rice K.L., (1987) [33]	Daily VAS, 57 ± 16 vs. 60 ± 11 Oxygen cost, 53 ± 19 vs. 58 ± 13 (codeine vs. promethazine)	12 min walk distance, 529 ± 315 vs. 564 ± 296 (codeine vs. promethazine)
Woodcock A.A., (1982) [18]	oxygen cost diagram (higher is better) 42.6 ± 12.0 vs. 47.6 ± 7.1 vs. 42.5 ± 8.6 (60 mg vs. 30 mg vs. placebo) *p* < 0.01 between 30 mg and placebo	6 min walk distance 396 ± 111 vs. 378 ± 114 vs. 368 ± 93 (60 mg vs. 30 mg vs. placebo)

Opioid group vs. placebo group, or differences in the values of opioid group from those of placebo group, are shown, if not indicated. Mean ± standard deviation or median [interquartile range] is shown, if not indicated. The range of visual analog scale (VAS) is 0–100 mm. The range of numerical rating scale (NRS) is 0–10, if not indicated. mMRC, modified Medical Research Council; 95%CI, 95% confidence interval; EORTC-QLQ-C15, European Organization for Research and Treatment of Cancer—Quality of Life Questionnaire Core 15; 6MWT, 6 min walking test. Red-colored outcomes show statistically significant differences favoring opioids. Green-colored outcomes show statistically significant differences favoring placebo.

**Table 3 ijerph-19-04907-t003:** Randomized controlled studies regarding acute effects of systemic opioids for non-cancer chronic respiratory disease.

**Parallel Studies**				
**Study (N)**	**Criteria of Dyspnea**	**Drug**	**Dose**	**Exercise Tolerance**
Kronborg-White S., (2020) [29] N = 18 vs. 18	MRC ≥ 3	IR Morphine	5 mg	Change of distance in 6MWT, 12 ± 34 vs. 1 ± 34 Change of Borg scale after 6MWT, 0 [0,0] vs. 0 [0,1], *p* < 0.05
**Crossover Studies**				
**Study (N)**	**Criteria of Dyspnea**	**Drug**	**Dose**	**Exercise Tolerance et al.**
Abdallah S.J., (2017) [34] N = 23	mMRC ≥ 3	IR morphine	0.1 mg/kg BW to a max. 10 mg	Cycle exercise time (min), 8.6 ± 6.5 vs. 6.2 ± 4.4, *p* < 0.05 Breathlessness Intensity during isotime, 3.0 ± 1.6 vs. 4.2 ± 2.6 (Borg scale), *p* < 0.05 Breathlessness Unpleasantness during isotime 3.1 ± 1.7 vs. 4.5 ± 2.6 (Borg scale), *p* < 0.01
Light R.W., (1996) [19] N = 7	Exercise limited by breathlessness	Morphine	30 mg	Change after the treatment (No statistical comparison) Cycle ergometer workload (W), 0.7 ± 6.1 vs. 1.4 ± 6.9 Borg score at equivalent workload, −0.14 ± 0.09 vs. 0.14 ± 0.69
N = 9	Exercise limited by breathlessness	Morphine	30 mg	Change after the treatment Cycle ergometer workload (W), 11.1 ± 6.5 vs. 5.7 ± 7.9 (Morphine + promethazine vs. promethazine)
Eiser N., (1991) [31] N = 8	No	Diamorp-hine	7.5 mg	Distance in 6MWT, 272 ± 49 vs. 263 ± 51 VAS dyspnea score for 6MWT, 62 ± 7 vs. 61 ± 7
Light R.W., (1989) [35] N = 13	Exercise limited by breathlessness	Morphine	0.8 mg/kg BW, once	Borg score at rest 0.29 ± 0.58 vs. 0.13 ± 0.2 Cycle ergometer workload (W) 93.1 ± 34.3 vs. 78.5 ± 32.2, *p* < 0.001 Exercise duration (min), 7.54 ± 2.09 vs. 6.50 ± 2.05, *p* < 0.001 Borg score at equivalent workloads, 7.08 ± 2.35 vs. 8.59 ± 2.31, *p* < 0.001
Johnson M.A., (1983) [36] N = 19	MRC ≥ grade 3	Dihydro-codeine	15 mg before Exercise TID at the max. for 1 wk	Daily VAS 46 ± 21 vs. 56 ± 23, *p* = 0.001, Distance in the treadmill (m), 249 ± 139 vs. 213 ± 127, *p* < 0.01 VAS of breathlessness at the equal distance, 67 ± 23 vs. 76 ± 19, *p* < 0.001
Woodcock A.A., (1981) [37] N = 12	MRC ≥ grade 3	Dihydro-codeine	1 mg/kg	Distance in treadmill (m) 347 ± 107 vs. 308 ± 90, *p* < 0.05 VAS of breathlessness at the equal distance, 55.4 ± 19.1 vs. 63.3 ± 20, *p* < 0.05

The study by Kronborg-White, et al. [29] included patients with fibrotic interstitial lung diseases. Other studies included patients with chronic obstructive pulmonary disease. Opioid group vs. placebo group is shown. Mean ± standard deviation or median [interquartile range] is shown. The range of visual analog scale (VAS) is 0–100 mm. MRC, Medical Research Council; IR, immediate-release; max, maximum; BW, body weight; wk, week; 6MWT, 6 min walking test. Red-colored outcomes show statistically significant differences favoring opioids.

**Table 4 ijerph-19-04907-t004:** Characteristics of non-randomized or observational studies of systemic opioids for non-cancer chronic respiratory disease.

**Non-Randomized Studies**							
**Study**	**Diseases**	**N**	**Completion Rate (%)**	**Criteria of Dyspnea**	**Drug**	**Dose**	**Period**
Rocker G.M., (2013) [38]	COPD	44	73	MRC 4 or 5	IR morphine	Titration from 0.5 mg BID	4–6 mos
Currow D.C., (2011) [26]	COPD et al.	83	63	mMRC ≥ 3	Morphine	Titration from 10 mg to 30 mg at the max.	Weekly titration period
Allcroft P., (2013) [39]	COPD	11	90.9	mMRC ≥ 2	SR morphine and clonazepam	10 mg	4 days
Allen S., (2005) [40]	IPF	11	100	Dyspnea at rest	Diamorphi-ne	2.5 mg for BW ≤ 60 kg, 5 mg for BW > 60 kg, SCI	15 min and 30 min
**Observational Studies**							
**Study**	**Diseases**	**N**	**Criteria of Dyspnea**		**Drug**	**Period**	
Smallwood N., (2018) [41]	COPD et al.	74	No		Not determined		
Vicent L., (2017) [42]	Respiratory diseases or heart failure	258	No		Not determined	During hospitalization	

COPD, chronic obstructive pulmonary disease; IPF, idiopathic pulmonary fibrosis; MRC, Medical Research Council; mMRC, modified Medical Research Council; IR, immediate-release; SR, sustained-release; max, maximum; BW, body weight; SCI, subcutaneous injection; mos, months.

**Table 5 ijerph-19-04907-t005:** Outcomes of non-randomized or observational studies of systemic opioids for non-cancer chronic respiratory disease.

**Non-Randomized Studies**		
**Study**	**Percentage of Improvement**	**Scores of Breathlessness** **Quality of life**
Rocker G.M., (2013) [38]	61%, helpful in the enrolled participants	NRS of dyspnea, −2.0 [−3.0 to 1.0], *p* = 0.02 CRQ, 0.6 [0.1 to 1.3], *p* < 0.001 CRQ Dyspnea, 0.6 [0 to 1.4], *p* = 0.004 McGill Quality of Life Questionnaire 1.0 [0 to 2.0] in the completed patients
Currow D.C., (2011) [26]	63%, ≥10% benefit on VAS of breathlessness 51%, ≥15% benefit on VAS of breathlessness in the enrolled patients	VAS, −13.5 ± 18.5 in mMRC = 3 (*n* = 20) VAS, −5.2 ± 19.9 in mMRC = 4 (*n* = 63) VAS, −17.1 ± 11.6 in responders (*n* = 52)
Allcroft P., (2013) [39]	50%, >15% reduction on VAS of breathlessness	Median VAS right now (morning) 49.5 (range 6–87) vs. 68.5 (range 31–86) Median VAS right now (evening) 45.4 (range 2–84) vs. 63.5 (range 9–75)
Allen S., (2005) [40]		15 min, average VAS 36.0 ± 11.0 vs. 83.0 ± 13.0 *p* < 0.0001 30 min, average VAS 36.0 ± 12.0 vs. 83.0 ± 13.0
**Observational Studies**		
**Study**	**Outcome**	
Smallwood N., (2018) [41]	41.9% self-reported as being very compliant with morphine treatment	
Vicent L., (2017) [42]	The unique independent predictor of a larger decrease in dyspnea was opioid treatment (*p* = 0.028)	

After opioids vs. before opioids, or differences in the values after opioid administration from those before opioids, are shown. Mean ± standard deviation or median [interquartile range] is shown, if not indicated. The range of VAS, visual analog scale, is 0–100 mm. The range of NRS, numerical rating scale, is 0–10, if not indicated. CRQ, chronic respiratory questionnaire (higher scores mean better QoL).

**Table 6 ijerph-19-04907-t006:** Retrospective studies of systemic opioids for non-cancer chronic respiratory diseases.

**Study (N)**	**Criteria of Dyspnea**	**Drug**	**Dose**	**Period**	**Percentage of Improvement** **Scores of Breathlessness**
Takeyasu M., (2016) [43] (N = 22)	Before deathdue to AE-IP	Morphine	Median initial 0.4 mg/h CIV (range 0.2–0.8) Maximum 0.8 mg/h CIV (range 0.2–2.9)	Within 24 h	40.9%, good; 36.4%, moderate; 18.2%, poor.
Matsuda Y., (2017) [44] (N = 25)	NRS ≥ 3	Morphine	Median initial, 0.25 mg/h At 2 h, 0.25 mg/h At 4 h, 0.5 mg/h CSI	2 h and 4 h	NRS at 2 h 5.52 ± 2.43 vs. 7.08 ± 2.33, *p* = 0.11 NRS at 4 h (N = 21), 5.32 ± 2.58 vs. 7.08 ± 2.33, *p* = 0.04
Bajwah S, (2012) [5] (N = 22)	Before death	Not determined		Documentation of effectiveness 95%, effective; 5% no documentation	
Tsukuura H., (2013) [45] (N = 6)	Prognosis of <1 month	Morphine et al.	CIV et al.		modified Borg scale 3.9 ± 3.1 vs. 4.7± 2.1, *p* = 0.683
Colman R., (2015) [46] (N = 55)	Lung transplant candidates	Titration from morphine 2.5–5 mg oral, or hydromorphone 0.5–1 mg oral		ESAS (N = 38), 39%, improvement in dyspnea Treadmill test (N = 14), Intensity 1.96 METs vs. 1.89 METs Exertion 19.28 kcal vs. 16.97 kcal, *p* = 0.06	

The study by Colman et al. [46] included patients with interstitial lung disease (ILD) and other diseases. Other studies exclusively included patients with ILD. After opioid vs. before opioid is shown, if not indicated. AE-IP, acute exacerbation of interstitial pneumonia; NRS, numerical rating scale; CIV, continuous intravenous infusion; CSI, continuous subcutaneous infusion; h, hour; ESAS, Edmonton Symptom Assessment System.

**Table 7 ijerph-19-04907-t007:** Quality of life (QoL) or other comprehensive assessments in randomized controlled studies.

Study	Drug	Criteria of Dyspnea	Period	QoL or Other Comprehensive Assessment
Verberkt C.A. (2020) [11]	mMRC ≥ 2	SR morphine	4 wks	CAT −2.18 points (95%CI, −4.14 to −0.22)
	mMRC ≥ 3	SR morphine	4 wks	CAT −1.17 points (95%CI, −4.17 to 1.84)
Currow D., (2020) [12]	mMRC ≥ 2	SR morphine *	1 wk	QOL EORTC-QLQ-C15, 0.35 (95%CI, −4.41 to 5.11) Q1, 48.5% vs. 49.3%, Q2, 43.0% vs. 47.3% Daily doses of rescue morphine, −0.56 (95% CI −0.92 to −0.18)
Kronborg-White S., (2020) [29]	MRC ≥ 3	IR Morphine	1 wk	Change of KBILD score 2.9 ± 6.6 vs. 1.6 ± 6.4
Ferreira D.H., (2019) [21]	mMRC ≥ 3	Oxycodone *	1 wk	QOL EORTC-QLQ-C15, −4.48 (95%CI, −11.69 to 2.73) Q1, 44.9% vs. 51.3%, Q2, 27.3% vs. 50.6%, *p* = 0.006 Daily doses of rescue morphine, −0.61 (95%CI, −1.02 to −0.20)
Abdallah S.J., (2017) [34]	mMRC ≥ 3	IR morphine	1 day	Preference for morphine over placebo for exercise 75%, Preference for placebo over morphine for exercise 15%
Abernethy A.P., (2003) [25]	Dyspnea at rest	SR morphine	4 days	No difference in overall wellbeing Sleep disturbed by breathlessness, 13.2% vs. 31.6%, *p* = 0.039
Poole P.J., (1998) [30]	No	SR morphine	6 wks	CRQ Total 2.08 ± 16.9 vs. 2.94 ± 12.9, *p* = 0.95 CRQ Dyspnea 2.50 ± 5.3 vs. 0.44 ± 3.2, *p* = 0.15 CRQ Mastery −0.36 ± 4.5 vs. 2.51 ± 3.7, *p* = 0.02 CRQ Fatigue −0.71 ± 4.1 vs. 0.71 ± 2.4, *p* = 0.34 CRQ Emotional −0.07 ± 6.7 vs. −0.86 ± 7.6, *p* = 0.73
Eiser N., (1991) [31]	No Diamorphine		2 wks	No significant difference in VAS of wellbeing
Johnson M.A., (1983) [36]	MRC ≥ grade 3	Dihydro-codeine	1 wk	Pedometer distance (km) for 1 week, 10.9 ± 8.0 vs. 9.3 ± 7.6, *p* < 0.05 No difference in alternate day treatment

Opioid group vs. placebo group, or differences in the values of opioid group from those of placebo group, are shown, if not indicated. Mean ± standard deviation is shown, if not indicated. * Participants in the placebo group also received rescue morphine. Q1, I have been less breathless during the past week. Q2, This medication would benefit me enough to be on it long term. QoL, Quality of life; mMRC, modified Medical Research Council; MRC, Medical Research Council; IR, immediate-release; SR, sustained-release; wk, week; 95%CI, 95% confidence interval. CAT, COPD assessment test (higher scores mean poorer QoL); CRQ, chronic respiratory questionnaire (higher scores mean better QoL); EORTC-QLQ-C15, European Organization for Research and Treatment of Cancer—Quality of Life Questionnaire Core 15 (higher scores mean poorer QoL); KBILD score, King’s Brief Interstitial Lung Disease score (higher scores mean better QoL).

## Data Availability

Data sharing not applicable.

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
