# Peer review of "Opioid Prescription Method for Breathlessness Due to Non-Cancer Chronic Respiratory Diseases: A Systematic Review"

_ijerph, 2022, doi:10.3390/ijerph19084907_

Round 1
Reviewer 1 Report
as below
Authors aimed to characterize and determine percentage the effect of opioids on patients with breathlessness due to non-cancerous breathlessness due to respiratory disease.
Their review of literature suggested to significant improvement in RCT in COPD patients but variable effect in observational studies.
They selected 18 studies from 763 studies for the systemic review. These studies were mainly in COPD and ILD patients and included RCT and observational studies.
They found non-significant sustained improvement in patients with severe breathlessness.
It is poorly executed systemic review and adds a little to existing knowledge.
Author Response
Thank you for your comments. Certainly, our review did not reveal much novel findings. However, it is important to show what was consistently reported, and what was not consistently reported. To the best of our knowledge, previous systematic reviews did not evaluate the details of opioid dose and prescription methods like our manuscript.
In addition, opioids are currently prescribed for intractable breathlessness in non-cancer respiratory diseases without clear evidence established. Although most of our findings were based on a limited number of studies, they provide suggestions for better prescription strategies for opioids in clinical practice. We added these explanations in discussion, as follows.
"Opioids are currently prescribed for intractable breathlessness in non-cancer respiratory diseases without clear evidence established. Although most of our findings were based on a limited number of studies, they provide suggestions for better prescription strategies for opioids in clinical practice. We clearly presented the evidence of morphine or codeine for COPD patients with an mMRC ≥ 3, which can be a cutoff point to consider when initiating opioids. Although adverse events are still a major obstacle to opioid use, the consistent findings regarding the acute effects of opioids on exercise tolerance support the initiation of as-needed low-dose IR morphine or codeine by setting the maximum number of uses. A gradual increment of very low dose (< 10 mg of morphine) would reduce the withdrawal of opioid usage due to adverse events. Nearly 60% responded well to morphine, and the majority of these responders required only low-dose morphine, (approximately 10 mg). Although the reports of this response rate were affected by placebo effects, these findings could directly be applied in actual practice."
Reviewer 2 Report
I think it's a good review. I would only recommend clarifying the type of patient, severity,clinical situation, in which the prescription of these drugs is recommended,
perhaps in the introduction and conclusions.
Author Response
Thank you for your comment.
In introduction, we added the following sentences to clarify the type of patient, severity and clinical situation.
“In clinical practice, physicians and patients experience difficulty in palliating breathlessness when the chronic respiratory diseases, such as COPD and interstitial lung disease (ILD), advance to breathlessness corresponding to modified Medical Research Council (mMRC) score ≥ 3 or 4.”
In discussion, we also added the following paragraph. We added this paragraph conclusively in discussion, not in conclusion, responding to the comments of other reviewer.
“Opioids are currently prescribed for intractable breathlessness in non-cancer respiratory diseases without clear evidence established. Although most of our findings were based on a limited number of studies, they provide suggestions for better prescription strategies for opioids in clinical practice. We clearly presented the evidence of morphine or codeine for COPD patients with an mMRC ≥ 3, which can be a cutoff point to consider when initiating opioids. Although adverse events are still a major obstacle to opioid use, the consistent findings regarding the acute effects of opioids on exercise tolerance support the initiation of as-needed low-dose IR morphine or codeine by setting the maximum number of uses. A gradual increment of very low dosage (< 10 mg of morphine) would reduce the withdrawal of opioid usage due to adverse events. Nearly 60% responded well to morphine, and the majority of these responders required only low-dose morphine (approximately 10 mg). Although the reports of this response rate were affected by placebo effects, these findings could directly be applied in actual practice.
Reviewer 3 Report
Many thanks to the Editor for allowing me to evaluate this work.
- Overall, this is an interesting study that draws on existing literature examining the effect of opioid prescription method for breathlessness due to non-cancer chronic respiratory diseases It is timely and scientifically sound and properly written, following all the guidelines for publications of scientific articles and it is within the scope of the International Journal of Environmental Research and Public Health. Overall evaluation: presented work contains 18 typewritten pages, including 1 figure and 5 tables. The abstract is compendious, the methods used to review literature are properly selected and described, the results are described in a clear and understandable way, the conclusions drawn are correct and not hasty; in addition, the Authors pay attention to the limitations of the study. Furthermore, 50 items of literature, thematically related to the content of the paper are properly cited.
I have only minimal comments:
- The discussion needs poprawek w obecnej formie problem został potraktowany zbyt powierzchownie. Proponuję aby Autorzy dokładniej przestudiowali zgromadzone dane I podali pełniejsze wnioski które mogą zostać wykorzystane w terapii.
- In my opinion the paragraph introduction should contain more informations and introduce in more detail the issues of presented manuscript.
- The discussion needs some improvements. In present form the problem has been treated too superficially. I suggest that the authors study the collected data in more detail and provide more complete conclusions that can be used in therapy.
Author Response
- In my opinion the paragraph introduction should contain more informations and introduce in more detail the issues of presented manuscript.
Thank you for your comment. We added the following paragraph to show the issues of the manuscript in introduction.
“However, recent randomized controlled trials (RCTs) did not demonstrate any significant differences in many outcomes between opioids and placebo [11,12]. The inconsistency of these results could be a barrier to prescribing opioids for breathlessness in non-cancer diseases [13]. In Japan, opioids for dyspnea have not been approved by the national insurance system, possibly owing to the absence of clear evidence of their efficacy. To establish the evidence of opioids for breathlessness, the cause of these inconsistencies should be clarified systematically, and the subgroups in which many patients respond well to opioids should be demonstrated.
In clinical practice, physicians and patients experience difficulty in palliating breathlessness when the chronic respiratory diseases, such as COPD and interstitial lung disease (ILD), advance to breathlessness corresponding to modified Medical Research Council (mMRC) score ≥ 3 or 4. The expected efficacy of opioids should be revealed in these patients. Simultaneously, both physicians and patients with non-cancer diseases prioritize the prevention of adverse events that can affect daily activities or prognosis. The safest prescription methods should be pursued to break barriers initiating opioids for the palliation of breathlessness in non-cancer diseases."
- The discussion needs some improvements. In present form the problem has been treated too superficially. I suggest that the authors study the collected data in more detail and provide more complete conclusions that can be used in therapy.
Thank you for your comment. We added the following paragraph in discussion to show better prescription strategies of opioids more conclusively.
“Opioids are currently prescribed for intractable breathlessness in non-cancer respiratory diseases without clear evidence established. Although most of our findings were based on a limited number of studies, they provide suggestions for better prescription strategies for opioids in clinical practice. We clearly presented the evidence of morphine or codeine for COPD patients with an mMRC ≥ 3, which can be a cutoff point to consider when initiating opioids. Although adverse events are still a major obstacle to opioid use, the consistent findings regarding the acute effects of opioids on exercise tolerance support the initiation of as-needed low-dose IR morphine or codeine by setting the maximum number of uses. A gradual increment of very low dose (< 10 mg of morphine) would reduce the withdrawal of opioid usage due to adverse events. Nearly 60% responded well to morphine, and the majority of these responders required only low-dose morphine (approximately 10 mg). Although the reports of this response rate were affected by placebo effects, these findings could directly be applied in actual practice.”